# Trace Elements in Dental Enamel Can Be a Potential Factor of Advanced Tooth Wear

Elzbieta Zamojda [1], Karolina Orywal [2], Barbara Mroczko [2] and Teresa Sierpinska [1,*]

[1] Department of Prosthetic Dentistry, Medical University of Bialystok, 24 Sklodowskiej-Curie Str., 15-276 Bialystok, Poland

[2] Department of Biochemical Diagnostics, Medical University of Bialystok, 15A Waszyngtona Str., 15-269 Bialystok, Poland

* Correspondence: teresa.sierpinska@umb.edu.pl; Tel.: +48-857485768; Fax: +48-857447030

**Abstract:** Enamel is the outermost layer of the tooth and has no ability to regenerate after tooth eruption. Its mineral composition is likely to be responsible for the enamel resistance to any agents that appear in oral cavity. The objective of this study was to compare the content of Ca, Mg, Zn and Cu in specific layers of healthy and worn tooth enamel. Three groups of dental enamel samples were analyzed. The amounts of Ca, Mg and Zn in the enamel bioptates were established using atomic absorption spectroscopy after an acid biopsy technique had been applied. The concentration of Cu was established using an electrothermic method with an AA spectrometer. The analysis was carried out using parametric tests (the Pearson's linear correlation coefficient and the *t*-test for the correlation coefficient). Results: A statistically significant difference was confirmed between the mean amount of zinc in severely worn dentition and all the layers in vitro and in vivo. Strong positive relationships between magnesium and zinc contents were shown at some depths. A significant difference was registered between the amount of copper in the severely worn dentitions compared to the superficial layer of the healthy teeth in vitro and in vivo. Conclusion: It appears that zinc metabolism may play an important role in enamel formation and may influence enamel's resistance to wear after tooth eruption.

**Keywords:** calcium; magnesium; zinc; copper; tooth wear; dentistry





## 1. Introduction

Enamel is the outermost layer of the crown of the tooth and is highly mineralized [1]. It is one of the most important structures of the tooth, both from a functional and aesthetic point of view [2]. The enamel structure is developed during the process of odontogenesis [3]. The enamel chemical composition and structure are likely to be responsible for the enamel resistance to chemical, thermal and biological agents that may appear in the human mouth [4,5]. Calcium and magnesium are the main minerals of crystals (hydroxyapatites) that are essential for enamel's structure and resistance [1]. Magnesium ions play an important role in the regulation of hydroxyapatite crystal growth. It can be adsorbed on the crystal's surface or can be built in its hydration layer [6]. Microelements, particularly zinc and copper, actively participate in the process of transformations within the enamel matrix, demineralization and remineralization [7,8]. Those minerals have the ability to form stable complexes with proteins, owing to so called zinc fingers, and take part in enzyme activation or inhibition [9]. Zinc plays a significant role in the metabolism of multiple factors and proteins, such as kallikrein 4, alkaline phosphatase, Krox 25 and Krox 26 transcription factors [10], and enamelysin, which has the ability to bind zinc and calcium ions [11]. Zinc-containing hydroxyapatite solutions can block the effect of acid attack on enamel by adhering to the enamel surface and forming a protective layer in vivo, significantly reduce the loss of enamel microhardness caused by bleaching and keep the enamel morphology nearly unchanged [12]. Copper ions inhibit the dental plaque formation, acid solubility

of enamel and enamel remineralization [13,14]. Its activity consists of the inhibition of bacterial growth and the activity of bacterial enzymes as a result of the oxidation of their key thiol groups [15]. However, little is known about the role of copper in enamel formation.

Correctly formed tooth enamel is the most resistant tissue in the human body [16]. However, in many cases, the process of severe wear may affect the structure of teeth. There are a lot of theories explaining why the pathological process of tooth wear occurs [7,17,18]. The cause of pathological tooth wear is multifactorial, and it is often difficult to determine the primary etiology. Some specific etiological factors that may play a key role are excessive occlusal forces, bruxism, premature contacts in function, eccentric loading of the teeth and hyperactivity of the masticatory muscles. Anyway, the hypothesis that some unknown processes occurring during enamel formation accelerate tooth wear appears to be worthy to assess [13,17,19,20].

The objective of this study was to compare the content of calcium, magnesium, zinc and copper in specific layers of healthy and worn tooth enamel.

## 2. Material and Methods

To assess calcium, magnesium, zinc and copper amounts, three groups of dental enamel samples were analyzed: 50 from subjects with worn dentition (4 bioptates per one person; total numbers of bioptates: 200), 20 from healthy volunteers in vivo (4 bioptates per one person; total numbers of bioptates: 80) and 15 healthy teeth cut longitudinally into seven layers in vitro (4 bioptates per one layer; total numbers of bioptates: 140).

### 2.1. Samples

(1) A total of 50 enamel samples were taken from upper central incisors derived from patients with advanced tooth wear. The greatest destruction of tooth hard tissue was observed on occlusal surfaces and/or incisal margins, which is why the tooth wear index for these surfaces was used in the further comparative analysis. The mean value of tooth wear was $2.27 \pm 0.52$ according to the Smith and Knight index (TWI), and mean patient age was $49.5 \pm 9$ years [21]. To include patients in the study, the following criteria were applied: visible features of advanced tooth wear on teeth, no dental caries or periodontal disease, no conservative treatment and preventive professional application of fluoride in a dental clinic prior to recruitment to the study. The patients were referred to the Department of Prosthetic Dentistry due to a clinically apparent decrease in occlusal vertical dimension (more than 4 mm) and a consequent reduced self-esteem of face aesthetics that required to be restored.

(2) A total of 20 enamel samples were taken from upper central incisors without signs of pathological tooth wear from healthy volunteers aged $48.5 \pm 6$ years. They were asked to participate voluntarily in the study, as they presented at the Department to complete a prosthetic procedure relating only to a single tooth (e.g., crown or inlay).

(3) A total of 15 permanent human central upper incisors with completed formation and without any visible pathological changes (donors between 18 and 21 years of age, who expressed their written informed consent for using their extracted teeth for studies) were used in the study. Mechanical damage in the area of alveolar process or changes in periodontium were an indication for tooth extraction. All teeth used in this research were obtained from the Bank of Teeth, University of Bern, School of Dental Medicine, Department of Preventive, Restorative and Pediatric Dentistry, Switzerland. They were prepared for the study in accordance with ISO/TS 11405:2015 [22].

### 2.2. Ethical Approval

The clinical data were conformed to the criteria of the Helsinki Declaration, ICH Guideline for Good Clinical Practice. This protocol was approved by the Local Ethical Committee with an approval number of R-I-003/6/2006. Informed consent was obtained from each participant at the beginning of the study prior to confirmation of their eligibility

for the study. The participants were able to withdraw from the study at any time and for any reason without prejudice.

*2.3. Study Design*

The study was conducted according to the scheme:

(1)  Teeth preparation and acid biopsy in vivo;
(2)  Teeth preparation and acid biopsy in vitro;
(3)  Biochemical analysis of samples using the AAS (atomic absorption spectrometry) method.

### 2.3.1. Clinical Procedure for Tooth Wear Patients and Volunteers

The acid biopsy technique was applied to assess the Ca, Mg, Zn and Cu contents in the tooth enamel [23]. The enamel of the labial surface of the maxillary central incisors was cleaned with pumice (CleanPolish, Kerr, Scafari, Italy) (30 s), rinsed by distilled water (1 min) and dried by air (5 min). Four analytical-grade filter paper discs were placed in the middle part of the prepared surface. Next, 1 µL of 0.1 mol/1 perchloric acid solution ($HClO_4$) (prepared directly in the Department of Biochemical Diagnostics, Medical University of Bialystok, Poland) was pipetted directly onto the middle of each of these discs for 60 s. The acid was transferred using a micropipette (Eppendorf Varipipette 4710, Eppendorf-Nethler-Hinz, Hamburg, Germany). The acid was allowed to work on the enamel for 60 s. Immediately after removing the filter paper discs, the biopsy area was rinsed with distilled water and dried. Fluoride gel was applied to the enamel to protect the place of biopsy. The bioptates were transferred to 1.5 mL sterilized, capped tubes (Safe-Lock, Eppendorf, Hamburg, Germany). One well-qualified individual performed all of the biopsies.

### 2.3.2. Extracted Teeth Preparation

Longitudinal cuts in the central part of the labial surface of extracted teeth were conducted using a Microm HM 355 S instrument (Microm, International GmbH, Walldorf, Germany) to obtain enamel layers of 150 µm. Due to the high hardness of enamel, the dental tissue was cut at the speed of 1 mm/s. The microtome allows cutting with thickness of 0.5 µm to 150 µm and at the speed of 0 to 430 mm/s. The cutting plane and distance between successive cuts were determined on the basis of the location of and distance between the striae of Retzius. Finally, seven successive enamel layers were obtained: 0—(0–150 µm), 1—(150–300 µm), 2—(300–450 µm), 3—(450–600 µm), 4—(600–750 µm), 5—(750–900 µm) and 6—(900–1050 µm). The procedure of acid biopsy was utilized on each enamel layer. Biopsy specimens were marked with letters from A to O (designation of the study tooth) and assigned a successive number corresponding to a depth of the study layer.

### 2.3.3. Biochemical Analysis

Before biochemical analysis, bioptates were mineralized using microwave mineralization (Uni Clever II, Plazmatronika, Wroclaw, Poland). This method was used to completely degrade organic matter and convert it into inorganic substances. The amounts of Ca, Mg and Zn in the enamel bioptates were established using atomic absorption (AA) spectroscopy with an air/acetylene flame (Z-500), Hitachi, Tokyo, Japan) [23]. The concentration of each element was calculated using a calibration curve, and the curve for each element was constructed using the instrument. The concentration of Cu was established using an electrothermic method with argon gas on the AA spectrometer, as calculated from the appropriate calibration curve. The cut-off points for the methods used were: 0.31 mg/L for Ca, 0.017 mg/L for Mg, 0.011 mg/L for Zn and 0.42 µg/L for Cu.

The reproducibility and reliability of the method was made by the recovery method and estimation of the value of RSD (relative standard deviation). The recovery figures for the analytical methods used in estimation of Ca, Mg, Zn and Cu were 101%, 98.7%, 104% and 100.5%, respectively. The precision values of the methods for determination of Ca, Mg, Zn and Cu were 3.4%, 3.6%, 2.6% and 2.8%, respectively.

*2.4. Statistical Analysis*

　　　The statistical description of individual characteristics was conducted. The study variables were quantitative in nature, and arithmetic mean and standard deviation were provided for those variables. The distribution of parameters was normal and in accordance with the Shapiro–Wilk test; therefore, the analysis was carried out using parametric tests. The strength of relationships between the pairs of study parameters was measured using the Pearson's linear correlation coefficient, and its significance was evaluated using the *t*-test for the correlation coefficient. The results for which the *p*-value was <0.05 were considered to be statistically significant.

　　　The statistical analysis of results obtained was performed using Statistica 10.0. (StatSoft PL. Krakow, Poland).

## 3. Results

　　　Mean values and standard deviations (SD) of calcium, magnesium, zinc and copper for worn dentition, healthy teeth in vivo and particular layers of healthy teeth in vitro are presented in the Table 1. It is worthy to note that mean values of calcium (Ca) and magnesium (Mg) did not differ between all the analyzed specimens, and a statistical difference was not found between the superficial layer of healthy teeth and worn dentition in vivo (Ca: $1.88 \pm 1.38$ mg/L vs. $1.85 \pm 1.24$ mg/L, ns; Mg: $0.30 \pm 0.14$ mg/L vs. $0.33 \pm 0.15$ mg/L, ns) and particular layers of enamel in vitro for calcium and magnesium amounts. However, the most interesting finding was revealed for zinc amounts. The mean value of zinc was two-times higher in the samples obtained from worn teeth compared to the healthy teeth ($0.14 \pm 0.04$ mg/L vs. $0.08 \pm 0.06$ mg/L, $p < 0.05$). A statistically significant difference was confirmed between mean value of zinc in severe worn dentition and all the layers, including a superficial one of healthy teeth, in vitro (00: $p < 0.001$, 0: $p < 0.001$, 1: $p < 0.001$, 2: $p < 0.001$, 3: $p < 0.001$, 4: $p < 0.001$, 5: $p < 0.001$, 6: $p < 0.001$) and in vivo ($p < 0.05$).

**Table 1.** The amounts of calcium (Ca), magnesium (Mg), zinc (Zn) and copper (Cu) in the study groups.

| Mineral | Worn Teeth (n = 50) | Healthy Teeth In Vivo (20) | 00 | 0 (0–150 μm) | 1 (150–300 μm) | 2 (300–450 μm) | 3 (450–600 μm) | 4 (600–750 μm) | 5 (750–900 μm) | 6 (900–1050 μm) |
|---|---|---|---|---|---|---|---|---|---|---|
| Ca [mg/L] | $1.88 \pm 1.38$ | $1.85 \pm 1.24$ | $1.42 \pm 0,39$ | $1.86 \pm 1.54$ | $1.59 \pm 1.22$ | $1.89 \pm 1.63$ | $1.96 \pm 0.86$ | $1.56 \pm 0.96$ | $1.47 \pm 1.39$ | $1.45 \pm 1.29$ |
| Mg [mg/L] | $0.30 \pm 0.14$ | $0.33 \pm 0.15$ | $0.18 \pm 0.08$ | $0.2 \pm 0.07$ | $0.31 \pm 0.19$ | $0.25 \pm 0.15$ | $0.26 \pm 0.16$ | $0.26 \pm 0.09$ | $0.31 \pm 0.13$ | $0.34 \pm 0.13$ |
| Zn [mg/L] | $0.14 \pm 0.04$ | $0.08 \pm 0.06$ [*] | $0.04 \pm 0.01$ [#] | $0.06 \pm 0.02$ [#] | $0.09 \pm 0.05$ [#] | $0.07 \pm 0.03$ [#] | $0.07 \pm 0.05$ [#] | $0.06 \pm 0.03$ [#] | $0.05 \pm 0.02$ [#] | $0.07 \pm 0.05$ [#] |
| Cu [μg/L] | $22.03 \pm 17.45$ | $36.67 \pm 22.66$ [*] | $10.42 \pm 5.56$ [#] | $17.87 \pm 6.59$ | $16.45 \pm 3.54$ | $16.22 \pm 8.63$ | $20.98 \pm 12.2$ | $17.65 \pm 7.71$ | $17.21 \pm 7.16$ | $14.55 \pm 4.27$ |

00, 0, 1, 2, 3, 4, 5, 6: particular layers of enamel in healthy extracted teeth. [*]: statistical significance of values $p < 0.05$ between worn teeth group and healthy teeth in vivo. [#]: statistical significance of values $p < 0.001$ between worn teeth group and particular layers of enamel in healthy extracted teeth in vitro.

　　　Relationships between the magnesium and zinc contents in individual enamel layers (in vitro) taking into consideration the Pearson's linear correlation coefficient and the significance level (*p*) for a particular correlation are presented in Figures 1–3. Statistically significant ($p < 0.05$) strong positive relationships between magnesium content and zinc contents at depths: 150–300 μm (Mg 1-Zn 1) (Figure 1); 450–600 μm (Mg 3-Zn 3) (Figure 2); and 900–1050 μm (Mg 6-Zn 6) (Figure 3) are shown.

　　　When analyzing copper amounts, few essential findings were revealed. The only significant difference was registered between the amount of copper in the severely worn dentitions compared to the superficial layer of the healthy teeth in vitro ($22.03 \pm 17.45$ μg/L vs. $10.42 \pm 5.56$ μg/L, $p < 0.01$) and in vivo ($22.03 \pm 17.45$ μg/L vs. $36.67 \pm 22.66$ μg/L, $p < 0.05$).

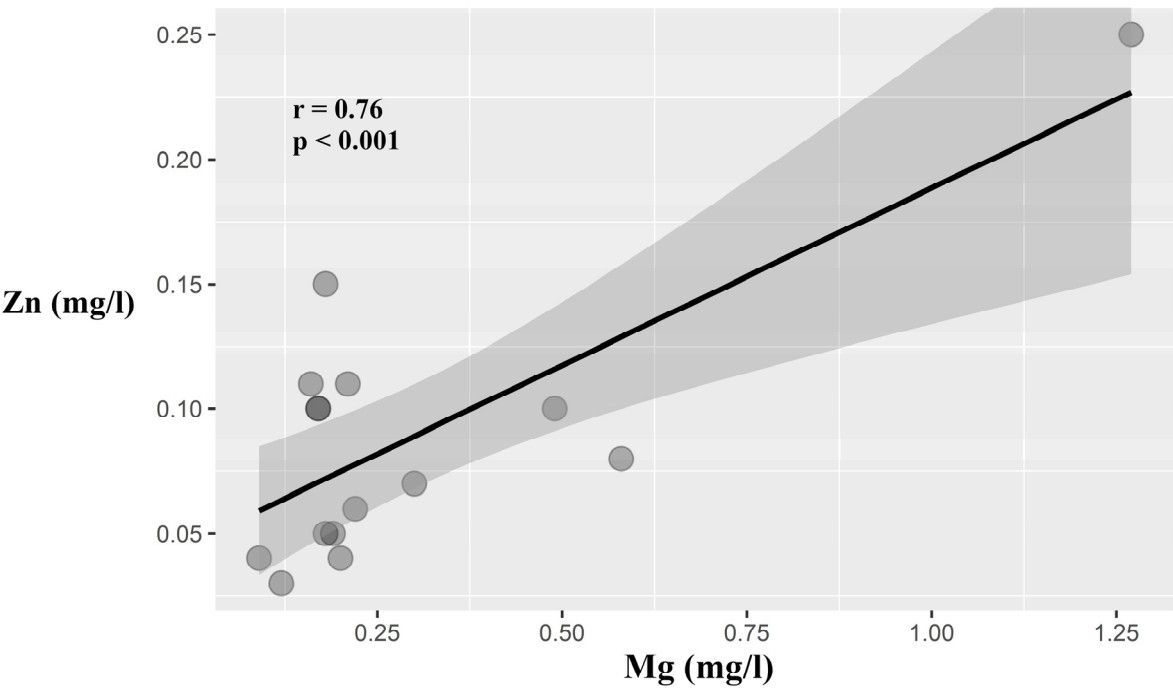

**Figure 1.** Scatter plot of correlation between Mg and Zn in first enamel layer.

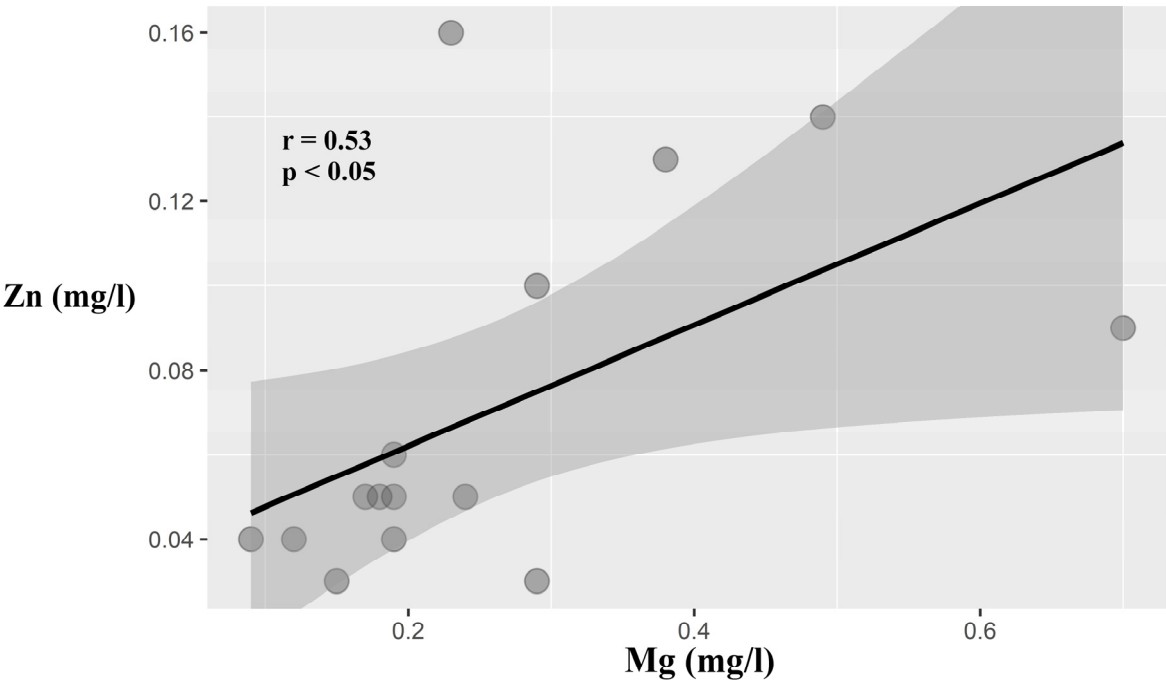

**Figure 2.** Scatter plot of correlation between Mg and Zn in third enamel layer.

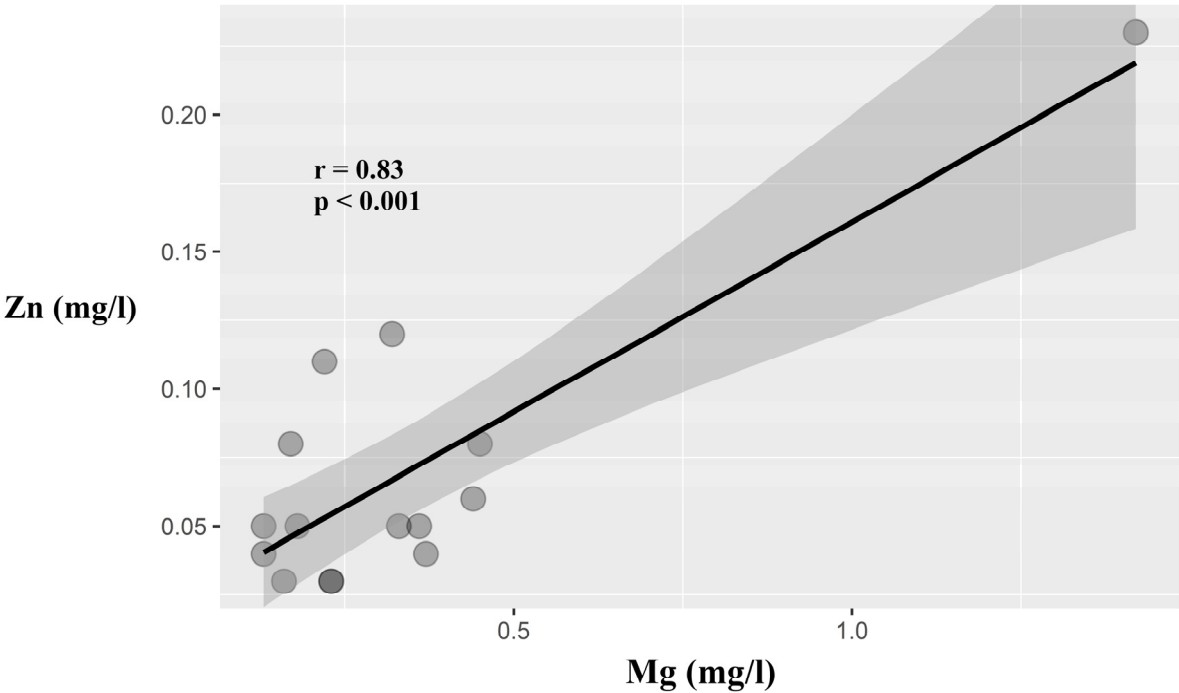

**Figure 3.** Scatter plot of correlation between Mg and Zn in sixth enamel layer.

## 4. Discussion

It is difficult to judge what the primary source of advanced tooth wear is and if tooth development may play any role in this process. It is also difficult to establish one single etiological factor of advanced enamel wear. The structure and composition of enamel as well as their environment may be regarded as determinants of the dental lesions [18].

Only the external layer of human enamel is accessible to investigate in vivo. However, it is most interesting is to assess deeper layers of enamel that are formed before tooth eruption. It was the reason to choose the method of acid biopsy. Acid biopsy is an invasive method but has avoided the need for invasive histological preparations that would have required tooth extraction. Animal studies tried to explain many of the controversies, but the human model may differ from the animal one. Enamel is made up of inorganic and organic parts; therefore, a decision was made to analyze the relationship of calcium and magnesium macroelements (the main enamel-forming minerals) on zinc and copper microelements (related to the enamel formation and changes). Macro- and microelements contained in the inorganic part of the tooth should occur at strictly defined concentrations [8] because, owing to them, multiple metabolic processes are initiated in the dental tissue during the individual development. Disturbed proportions or the absence of some elements can affect the content of other minerals and result in greater tooth vulnerability to dental caries and other pathological agents [8]. When analyzing relationships between calcium and zinc in the enamel, the enamel formation period before tooth eruption should be taken into consideration. An enzyme called enamelysin, which is capable of binding two $Ca^{2+}$ and two $Zn^{2+}$ ions, is the most active during the secretory stage of enamel formation [11]. The elevated zinc level in some enamel layers is associated with the high activity of enamelysin, which inhibits the transformation of amorphous calcium phosphate to hydroxyapatite crystals [24]. Zinc is also an inhibitor of KLK4, of which abnormal activity is associated with abnormal enamel maturation and its reduced calcification. Enamel has normal spatial organization in the absence of KLK 4; however, its mineralization decreases with increasing depth [25]. A conclusion could be drawn on this basis that zinc content increases with decreasing calcium content (lower enamel calcification). The testing performed as part of the study has not confirmed it, which may result from the study teeth contact with the oral cavity environment for many years. According to some authors, the absorption of zinc by

enamel may be a part of posteruptive enamel maturation [26]. There are reports that zinc is capable of modifying the growth of hydroxyapatite crystals, of which the main component is calcium. The mechanism of $Zn^{2+}$ ion absorption on the hydroxyapatite crystal surface [27] affects the growth inhibition of not only hydroxyapatite crystals, but also their precursors, dicalcium phosphate dihydrate (DCPC) and octacalcium phosphate (OCP) [28]. Studies with synthetic hydroxyapatite showed that zinc can be incorporated into the hydroxyapatite crystal lattice, which leads to its increased resistance to acid [1]. During this process, the central calcium atom is replaced by zinc [12]. Therefore, zinc competes with calcium for positions on the apatite crystal surface and can be easily displaced by calcium from hydroxyapatite [29]. Calcium can also reduce zinc adsorption by hydroxyapatite precursors, and this effect is potentially intensified by the increased number of growth centers which are not occupied by zinc [30]. The above reports suggest that the zinc content increases with the decreasing calcium content in hydroxyapatite because calcium is displaced by zinc. However, this hypothesis has not been confirmed by the conducted study.

Metabolism of magnesium in human enamel is not well-studied. As far as it is known, magnesium is present in the enamel as magnesium phosphate. It may also influence the activity of alkaline phosphatase and may catalyze the process of properly shaped hydroxyapatite crystal development. Magnesium may also inhibit the transition of calcium phosphate from noncrystallized into crystallized form [31,32]. It has also been reported that magnesium can be easily replaced by zinc in biological systems; therefore, the buffered concentration of zinc is held at least a million-fold below the magnesium content inside the majority of cells [29]. Such a relationship may also apply to the tooth enamel in the formation period. In studies on rats, it has also been proven that an inappropriate diet fed to pregnant mothers affects the zinc and magnesium levels in dental hard tissues in 2-month-old offspring. A decrease in the content of those minerals in dental hard tissues in 2-month-old offspring has also been observed with decreasing zinc and magnesium contents in the mother's diet [33]. Attention should be drawn to the fact that correlations between magnesium and zinc are cyclical in nature in a way (every 150 μm), which can also be associated with the cyclical and phased enamel formation. This process is reflected by the presence of the striae of Retzius in the enamel structure [32]. It has been proven that zinc is essential for the normal enamel formation, and its deficiency means a greater tooth vulnerability to dental caries. However, the presence of abnormal enamel areas has been noted in studies on children's teeth after cancer treatment (chemotherapy). In those patients, the enamel showed higher magnesium and zinc levels compared to the control group [34]. It can be concluded on this basis that the zinc content increases with the increasing magnesium content, and the enamel resistance to pathological factors is reduced. The results obtained in the study demonstrated two-times-higher mean value of zinc in the worn dentitions when compared to healthy teeth.

Few reports about interactions between calcium and copper in the tooth enamel have been found. It is believed that the mechanism of copper ion action after the tooth eruption consists in the creation of an insoluble copper phosphate layer on the tooth surface. The demineralization is reduced and inhibited by the precipitation of a protective copper phosphate phase through the crystal lattice stabilization on the enamel surface [14]. A conclusion can be drawn on this basis that the enamel calcium content increases with the increasing copper level in the solution surrounding the tooth, as its solubility is reduced. Animal studies suggest that Cu deficiency is associated with reduced bone strength and deterioration of bone quality, leading to osteoporotic lesions [35]. Matrix proteinase 20 cleaves aggrecan, cartilage oligometric matrix protein, type V collagen, type XVIII collagen, fibronectin and any other proteins [3]. Procollagen N-proteinase is engaged in the extracellular metabolism of the enamel matrix. It can be inactivated to 50% by copper when its amount is in the range of 14–40 μM. Additionally, glucosyl-transferase is inactivated by copper, and gelatinase A and B are inactivated by both zinc and copper [36].

Summarizing, it was found that the amount of zinc was considerably different when comparing worn dentition and healthy teeth. Even if it is very speculative, it appears that

zinc metabolism during enamel formation and maturation may influence the future enamel resistance to wear.

Additionally, recent research showed that some treatments can alter enamel composition and behavior, such as fluorides [37], casein phosphopeptide-amorphous calcium phosphate [38] and the recently introduced biomimetic hydroxyapatite [39].

A limitation of the study is that only four elements were analyzed. Therefore, other potential variables should be tested in future in vitro and in vivo investigations. Additionally, it should be taken into consideration that the content of a particular mineral in a biopsy specimen may indicate a low enamel vulnerability to dissolution under the influence of acid applied, or it may indicate the actual mineral content in enamel. Unfortunately, it is not possible to prepare typical histologic specimens of tooth enamel because the decalcification of enamel to prepare thin fragments makes it vulnerable to dissolving.

Thus, it is necessary to continue the research to explain the role not only of zinc in the process of wear, but any other potential agents.

**Author Contributions:** E.Z. and T.S. conceived and planned the study. E.Z. and K.O. carried out the study and took the lead in writing the manuscript. B.M. made statistics and prepared graphs. T.S. contributed to the interpretation of the results and supervised the protocol. All authors have read and agreed to the published version of the manuscript.

**Funding:** This research received no external funding.

**Data Availability Statement:** All the data are available in the Department of Prosthetic Dentistry, Medical University of Bialystok, Poland.

**Conflicts of Interest:** The authors declare no conflict of interest.

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
