# Peer review of "Trace Elements in Dental Enamel Can Be a Potential Factor of Advanced Tooth Wear"

_minerals, doi:10.3390/min13010125_

Round 1
Reviewer 1 Report
Dear Authors,
I have read the manuscript with interest and some questions raised. Enlisted please find my comments.
Overall. General English grammar revision (Minor spelling errors).
Key words. “dentistry” could be added in my opinion.
Abstract. Please add the names of the statistical tests in this section.
Introduction. Authors stated “There are a lot of theories explaining why the pathological process of tooth wear occurs. The cause of pathological tooth wear is multifactorial and it is often difficult to determine the primary etiology.”. Please add a reference for this statement.
Materials and Methods. Authors stated “50 from subjects with worn dentition and 20 from healthy 62 volunteers in vivo and 15 healthy teeth cut longitudinally into seven layers in vitro.”. Please add if and how sample size calculation has been performed.
Materials and Methods. Authors stated “The enamel of the labial surface of the maxillary central incisors 104 was cleaned with pumice, rinsed and dried”. Please add details concerning treatment time and materials used.
Materials and Methods. Authors stated “Next, 1 μL of 0.1 mol/1 perchloric acid 106 solution (HClO4) was pipetted […]The acid 107 was transferred using a micropipette (Eppendorf Varipipette 4710, Eppendorf-Nethler-Hinz, Germany) […]The bioptates were 111 transferred to 1.5 mL sterilized, capped tubes (Safe-Lock, Eppendorf, Germany).”. For each material used please add details about commercial name manufacturer, City and State.
Materials and Methods. Authors stated “Longitudinal cuts in the central part of the labial surface of extracted teeth were done 115 using a MICROM HM355S microtome, International GmbH. […]Before biochemical analysis bioptates were mineralized using microwave minerali-127 zation (Uni Clever II, Plazmatronika, Poland) […]The amounts of Ca, Mg and 129 Zn in the enamel bioptates were established using atomic absorption (AA) spectroscopy 130 with an air/acetylene flame (Hitachi Model Z-500, Spectro, Germany)”. For each machinery used please add details about commercial name manufacturer, City and State.
Materials and Methods. Authors stated “Due to the high hardness of enamel, the dental tissue was cut at the speed of 1mm/s.”. Please add a reference for this method.
Materials and Methods. Authors stated “Reproducibility and reliability agreement of the methods used were found to be 90%.”. Please explain how reproducibility was measured.
Materials and Methods. Authors stated “The distribution of parameters was normal”. How normality was tested. Please point out test used.
Materials and Methods. Authors stated “The statistical analysis of results obtained was performed using Statistica 10.0., StatSoft PL.”. Please add details about software used, version, Manufacturer, City and State.
Discussion. Authors stated “Summarizing, it was found that zinc amount was considerably different when com-268 paring worn dentition and healthy teeth. Even if, it is very speculative but it appears that zinc metabolism during enamel formation and maturation may influence the future enamel resistance to wear.”. Provide a general interpretation of the results in the context of other evidence, and implications for future research. It could be added that “Additionally recent research showed that some treatments can alter enamel composition and behavior, such as Fluorides (P Zampetti P, Scribante A. Historical and bibliometric notes on the use of fluoride in caries prevention. Eur J pediatr Dent 2020 Jun;21(2):148-152), casein phosphopeptide-amorphous calcium phosphate (Quantitative evaluation of remineralizing potential of three agents on artificially demineralized human enamel using scanning electron microscopy imaging and energy-dispersive analytical X-ray element analysis: An in vitro study. Khanduri N, Kurup D, Mitra M. Dent Res J (Isfahan). 2020 Sep 7;17(5):366-372.) and the recently introduced Biomimetic Hydroxyapatite (Butera A, Pascadopoli M, Gallo S, Lelli M, Tarterini F, Giglia F, et al. SEM/EDS Evaluation of the Mineral Deposition on a Polymeric Composite Resin of a Toothpaste Containing Biomimetic Zn-Carbonate Hydroxyapatite (microRepair®) in Oral Environment: A Randomized Clinical Trial. Polymers. 2021; 13(16):2740). Also these variables should be tested in future in vitro and clinical investigations”. These concerns should be added to discussion.
Discussion. Please add a paragraph showing the limitations of the present report.
References. Some references are quite old (1980;1984;1994;1996;1994;1997;1996;1998;1997). If possible, please switch with some more modern research. Some recent studies have been suggested in the sections above.
Author Response
Please see the attatchment

Reviewer 2 Report
The authors need to read the comments

Author Response
Please see the attachment.
The comment of the review is presented directly in the pdf delivered by Reviewer 2.

Round 2
Reviewer 1 Report
All comments have been answered. Thank you